# Role of GARP Vesicle Tethering Complex in Golgi Physiology

**DOI:** 10.3390/ijms24076069

**Published:** 2023-03-23

**Authors:** Amrita Khakurel, Vladimir V. Lupashin

**Affiliations:** Department of Physiology and Cell Biology, University of Arkansas for Medical Sciences, Little Rock, AR 72205, USA

**Keywords:** GARP complex, Golgi, vesicle tethering, SNARE, glycosylation

## Abstract

The Golgi associated retrograde protein complex (GARP) is an evolutionarily conserved component of Golgi membrane trafficking machinery that belongs to the Complexes Associated with Tethering Containing Helical Rods (CATCHR) family. Like other multisubunit tethering complexes such as COG, Dsl1, and Exocyst, the GARP is believed to function by tethering and promoting fusion of the endosome-derived small trafficking intermediate. However, even twenty years after its discovery, the exact structure and the functions of GARP are still an enigma. Recent studies revealed novel roles for GARP in Golgi physiology and identified human patients with mutations in GARP subunits. In this review, we summarized our knowledge of the structure of the GARP complex, its protein partners, GARP functions related to Golgi physiology, as well as cellular defects associated with the dysfunction of GARP subunits.

## 1. History and Discovery

The genetic and biochemical analyses of the protein traffic in yeast *S. cerevisiae* led to the identification of the Golgi Associated Retrograde Protein (GARP) complex. The goal of the original study was to discover new genes involved in protein trafficking between Golgi and pre-vacuolar (endosomal) compartments. A transposon-based mutagenic procedure was employed to create vacuolar protein sorting (vps) mutants [1]. All vps mutants missorted (secreted) vacuolar enzyme carboxypeptidase Y (CPY). This study identified three novel VPS mutants named *vps52*, *vps53*, and *vps54* involved in protein trafficking between Golgi and pre-vacuolar (endosomal) compartments. In comparison to wild type (WT) cells, these mutants grew slowly at 30 °C. Moreover, *vps52-54* mutants secreted a significant amount of CPY and had fragmented vacuolar morphology [2]. In addition, mutants were also sensitive to elevated concentrations of Zn^2+^, Mn^2+^, and Cd^2+^ ions and extreme pH [2]. The *vps54* mutant accumulated numerous small acidic vesicles. Proteins encoded by VPS52, 53, and 54 genes were thought to form a complex because both the triple *vps52/53/54* mutants and single *vps52*, *vps53*, *vps54* mutants do not have a significant difference in phenotype. Further analysis by immunoprecipitation of individual Vps52p, Vps53p, and Vps54p under non-denaturing conditions confirmed that these three proteins form a discrete complex. Moreover, Vps52p, Vps53p, and Vps54p were unstable and quickly degraded in the absence of the other two subunits [1]. The Vps52p/53p/54p complex was initially named Vps fifty-three (VFT) [3], which later was renamed GARP after the discovery of the fourth subunit, Vps51p [4,5]. This newly identified subunit of the complex was equally crucial to the membrane trafficking as other subunits. The yeast cells that lack Vps51p also have defects in vacuole morphology and recycling of SNARE Snc1p to the plasma membrane [4]. Similarly, in human cells, VPS51 knockdown (KD) partially dispersed the endosome to *trans*-Golgi network (TGN) recycling proteins CI-MPR and TGN46. The uptaken Shiga toxin did not reach the Golgi in VPS51 KD cells but scattered throughout the cytoplasm [6]. This indicates that Vps51 has a similar function to that of the VFT complex. At the same time, the assembly of the Vps52p/53p/54p complex can still take place in the *vps51* mutant, even though the level of these subunits was highly reduced compared to WT cells [1,3].

Even though the GARP complex was initially discovered in yeast, subunits of the GARP complex are evolutionarily conserved and have homologs in worms, flies, mammals, and plants [7,8,9].

## 2. Composition and Structure of the GARP Complex

The GARP complex belongs to the group of multisubunit tethering complexes (MTC) and is peripherally associated with the cytosolic face of TGN. It is a heteromeric protein complex composed of four different subunits-VPS51, VPS52, VPS53, and VPS54 [8]. In most eukaryotes, the length of the GARP complex subunits ranges from 650 to 1700 amino acid residues whereas the yeast Vps51p subunit is an exception as it is only 164 amino acids long [10].

The amino acid sequence in the *N*-terminus of every identified subunit of the GARP complex is predicted to form an alpha-helical coiled-coil motif required for the complex’s assembly. Hence, the complex is expected to have a core of amino terminals and four projecting arms corresponding to the subunits’ carboxy-terminal regions (Figure 1). The GARP subunits form an obligatory 1:1:1:1 complex [1,3,6].

In yeast, negative-stain electron microscopy of purified complex identified the structure of GARP as a Y-shaped complex with projecting legs labeled A–C. Leg A is the shortest, while leg C is the longest. A hook was present at the end of leg B, and a hinge was present in the middle of leg C (Figure 1A). To understand which leg each subunit corresponds to, double-GFP tagging was performed. Legs A, B and C were found to correspond to Vps52p, Vps53p and Vps54p, respectively. Vps51p was comparatively shorter and located at the connection point between the three legs of a Y-shaped structure [11].

In higher eukaryotes, in addition to GARP, the endolysosomal traffic is also regulated by the Endosomal Associated Retrograde Protein (EARP) complex. EARP is composed of VPS50/Syndentin, VPS51, VPS52 and VPS53 subunits. Three of its subunits, VPS51, VPS52, and VPS53, are shared with the GARP complex. EARP is involved in the recycling of a subset of plasma membrane receptors, such as transferrin receptors, from late endosomes back to the cell surface [12]. It is also essential for dense-core vesicle biogenesis [13].

The predicted 3D structure of the human GARP complex is strikingly similar to the appearance of yeast GARP as seen by negative-stain electron microscopy (Figure 1A,B). In both the yeast and human GARP complex, extended *C*-terminal “arms” are widely (up to 35 nm) open, allowing them to assemble docking platforms to tether 50–60 nm-wide trafficking intermediates (Figure 1). This predicted structure is likely to represent the “open” conformation of GARP.

The complete 3D structure of human GARP subunits are not available yet. However, in yeast, analysis of the crystal structure of the Vps53p *C*-terminus (564–799 residues) at a resolution of 2.9 A, revealed two alpha-helical bundles arranged in tandem [14]. The *C*-terminus of Vps53p consists of a highly conserved surface patch that may take part in vesicle recognition [14]. Interestingly, the *C*-terminal region of VPS53 is predicted to contain the so-called MUN domain [15]. This domain was initially found in UNC13/Munc13-1 protein and predicted in many subunits of the CATCHR vesicle tether family (SEC6, EXO70, COG4, and RINT1). Deletion of Munc13-1 in mice causes severe defects in neurotransmitter release, indicating its fundamental role in synaptic transmission [16]. It is important to note that the MUN domain is the minimal module responsible for Munc13-1/UNC13 activity [17].

The crystal structure of mouse VPS54 *C*-terminal 142 amino acid residues (836–977) at 1.7 A resolution, in conjunction with comparative sequence analysis, also revealed an alpha-helical bundle that is strikingly similar to domains found in other vesicle tethering complexes such as Dsl1, COG, and exocyst. It also revealed that Leu967 is buried within the alpha-helical bundle through predominantly hydrophobic interactions critical for domain stability and folding in vitro. Mutation of this residue to glutamine in wobbler mice (see Section 6.1.2.) dramatically reduces the half-life and total protein level of VPS54, even though the integration of the affected subunit into the GARP complex is not prevented [18].

The GARP complex belongs to the family of complexes associated with tethering containing helical rods (CATCHR) [11,14]. The comparison of the GARP complex with other tethering complexes of the CATCHR family uncovered a similarity in the subunit organization between GARP, COG, exocyst, and Dsl1 protein complexes. Comparisons of GARP complex with COG complex demonstrated that Vps51p structurally corresponds to Cog2p, Vps52p to Cog3p, Vps53p to Cog4p/Sec6/Dsl1, Vps54p to Cog1p [11,14,19].

**Figure 1 ijms-24-06069-f001:**
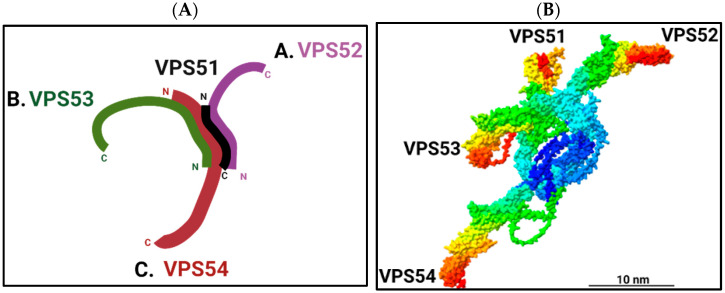
Structure of the GARP complex. (**A**) Schematic representation of the proposed 3D structure of yeast GARP complex based on negative-stain electron microscopy [11]. (**B**) AlphaFold structure prediction of human GARP complex. The *N*-terminal regions of VPS51 (1–194), VPS52 (1–178), VPS53 (1–169), and VPS54 (1–310) were used to assemble a 3D core structure of the GARP complex using the ColabFold [20] plugin of the UCSF ChimeraX program [21]. AlphaFold predicted 3D structures of individual subunits were fitted to the core structure using the “matchmaker” command of the UCSF ChimeraX program, and the resulting structures were colored using the Rainbow palette. Scale bar is 10 nm. Predicted 3D structures of individual VPS51-54 human proteins were updated in AlphaFold DB version 2022-11-01, created with the AlphaFold Monomer v2.0 pipeline. The predicted 3D structure of the GARP complex was assembled on 27 December 2022 and colored using ChimeraX.

## 3. Localization of the GARP Complex to the Golgi Membrane

The GARP complex is localized to *trans*-Golgi/TGN in both yeast and mammalian cells (Figure 2) [1,22,23]. In yeast, the recruitment of GARP was thought to be achieved by the Rab GTPase Ypt6p, which is activated by the guanidine exchange factor (GEF) Ric1p-Rgp1p [24]. Activation of Ypt6p mediates the tethering of vesicles containing SNARE protein Tlg1p to the TGN through GARP [24]. In human cells, the Ypt6p homolog RAB6 also interacts with VPS52. Still, this interaction is not essential for the association of the GARP complex with TGN since knockout (KO) of RAB6 did not abrogate the GARP localization [25]. Moreover, in mammalian cells, the recruitment of the GARP complex to TGN is primarily achieved by two small GTPases, ARL5 and ARFRP1. KO of ARL5 and ARFRP1 prevented the association of the GARP complex with Golgi, while expression of ARL5 restored the localization of the GARP complex, indicating the importance of these GTPases in the recruitment of the GARP complex. The activated ARL5-GTP associates with the TGN membrane through the myristoylated *N*-terminal domain [25]. It is currently not known which GARP subunit directly interacts with ARL5. The interaction of ARL5 and the GARP complex was also shown in *Drosophila melanogaster*. In both fly and human cells, deletion of ARL5 caused a partial displacement of GARP from the Golgi membranes and resulted in the accumulation of enlarged endosomal compartments [26].

Although VPS51, VPS52, and VPS53 are shared by both GARP and EARP complexes, the location of these two complexes differs. VPS54 colocalizes with the TGN protein p230/GOLGA4, while VPS50 of the EARP complex localizes to RAB4-containing endosomes demonstrating distinct intracellular localizations [12].

## 4. The GARP Complex Protein Partners

### 4.1. SNAREs

In *S. cerevisiae*, Vps51p interacts with Qc t-SNARE Tlg1p via the amino-terminal regulatory Habc domain of Tlg1p. The Tlg1p binding site was mapped to residues 1–30 in the *N*-terminus of the Vps51p molecule [3,4]. Further structural analysis of the *N*-terminal domain of Tlg1p bound to a peptide from the *N*-terminus of Vps51p revealed that Vps51p–Tlg1p binding depends on 18–30 residues of Vps51p. The binding specificity of the Vps51p–Tlg1p was validated by deletion analysis (residues 27–30) and point mutations (Phe27 to Leu or Tyr28 to Ala). The *N*-terminal Vps51p peptide is partially alpha-helical and is bound to a groove in a three-helix bundle formed by Tlg1p. However, the point mutation that alters the binding of Tlg1p to Vps51p did not show any trafficking defects indicating that Tlg1p–Vps51p interaction is dispensable for the function of the GARP complex [27]. Similarly, human VPS51 has also been shown to interact with the Habc domain of STX6, which is the human ortholog of Tlg1p, indicating that this protein–protein interaction is evolutionary conserved [6]. The crystal structure of the complex between the STX6 Habc domain and the *N*-terminus of VPS51 demonstrated that the contact region in the STX6 Habc domain corresponds to a hydrophobic groove that is highly conserved as compared to other parts of the surface. VPS51 is anchored to this groove by the burial of Tyr40 and Tyr41, and this indicates the requirement of the VPS51 di-tyrosine motif to interact with the Habc domain of STX6 and STX10 [27]. STX10 also pulled down human VPS52 and was indicated as a GARP interactor [23].

In human cells, in vitro and in vivo studies demonstrated that the GARP complex interacts with the STX16 TGN SNARE complex (STX16/STX6/VTI1A/VAMP4). Microscopy analysis demonstrated colocalization of VPS54 with STX16, STX6, and VAMP4. Importantly, siRNA-mediated KD of VPS54 showed a reduction in STX16 SNARE complex formation and resulted in the redistribution of the TGN SNAREs. These SNARE–GARP interactions are dependent on the SNARE motif in Golgi SNARE proteins and on *N*-terminal regions of VPS53 and VPS54 [28]. It is interesting to note that the “cousin” of GARP, the intra-Golgi vesicular tethering COG complex, interacts with Qa SNARE STX5 via the *N*-terminal domain of the COG4 subunit [29,30,31,32]. The exact nature of the GARP–SNARE interaction is not known, but it was shown that the assembled GARP complex could bind to the assembled STX16 SNARE complex [28]. It is important to note that the VPS53 protein contains a MUN structural domain [33] and that the founding member of the MUN family, Munc13-1, binds to both SNAREs and SNARE-interacting SM (Sec1/Munc18) proteins regulating the transition from Munc18-syntaxin-1 pre-fusion complex to a fusogenic *trans*-SNARE complex during synaptic vesicle fusion [34].

### 4.2. Small GTPases

In yeast, Vps52p interacts with the GTP-bound form of Rab GTPase Ypt6p. Similarly, mammalian RAB6 pulled down VPS52, indicating that there is an interaction between RAB6 and the GARP complex in mammalian cells [23]. Ypt6p is essential for retrograde transport from endosomes to the TGN, and its mutation can result in protein mislocalization and growth defects. Immunofluorescence imaging reveals that the depletion of Ypt6p disperses the GARP signal from TGN. However, the subcellular fractionation assays do not show a reduction in GARP association with the membrane fraction, indicating that Ypt6p is needed for proper intracellular localization but not for the GARP-membrane association [3,24]. Interestingly, another study showed that the defect in Ypt6p mutants, like endosome-TGN transport deficiency and growth delays, can be compensated by overexpression of either Arl1p or coiled-coil Golgi protein Imh1p, indicating that at least some of the GARP functions can be restored in a Ypt6-independent manner [35].

The Vps53p in *S. cerevisiae* interacts with the GTP-bound form of Arl1p [36]. However, the loss of Arl1p has no effect on GARP localization to the TGN [4,24]. Interestingly, in both mammals [25] and flies [26], GARP interacts with another GTPase of the ARL subfamily, ARL5, and this interaction is essential for TGN localization of the GARP complex.

In mammalian cells, VPS52 also interacts with ARF6, a small GTP-binding protein involved in endocytic trafficking. The interaction between ARF6 and VPS52 was demonstrated using the GTP-locked constitutively active ARF6 mutant (ARF6[Gln67Leu]). The *N*-terminal 98 amino acids of the VPS52 are not required for the binding to ARF6. In contrast, the deletion of *N*-terminal 149 amino acids or *C*-terminal 50 amino acids from VPS52ΔN48 (VPS52ΔN48/C50) disrupted the interaction with ARF6(Gln67Leu). This data indicates that either *N*-terminal amino acids 99–149 and *C*-terminal amino acids 674–723 or the ternary structure of VPS52, are required for the interaction with ARF6. While VPS52ΔN149, the mutant that lacks ARF6 interaction, was able to rescue some VPS52 KD phenotypes, it was not able to rescue the increase in axonal length of hippocampal neurons [37]. This indicates the importance of ARF6–VPS52 interaction in the regulation of neuronal morphogenesis.

### 4.3. Coiled-Coil Tethers

It was proposed that during ER stress, the interaction between Vps53p and golgin Imh1p is augmented to facilitate the recycling of yeast SNAREs Snc1p and Tlg1p to the late Golgi. Following ER stress, the unfolded protein response induces phosphorylation of Arl1p GEF Syt1p to recruit more Imh1p to the late Golgi. The membrane pool of Imh1p undergoes Ser25/Thr27 phosphorylation through MAPK Slt2/ERK2 activation. Imh1p, once phosphorylated, activates the GARP complex to enhance the recycling of Tlg1p-positive vesicles to the late Golgi, thereby facilitating the transport of Snc1 to the Golgi and plasma membrane. Consequently, under ER stress, two distinct tethers, coiled-coil Golgin Imh1p and the GARP, work together to promote the recycling of the Golgi SNAREs Snc1p and Tlg1p to the late Golgi [38].

### 4.4. Other Partners

#### 4.4.1. EIPR1/TSSC1

TSSC1 (tumor-suppressing subchromosomal transferable fragment candidate gene 1) is a 43.6 KD TGN-localized protein that is predicted to contain a WD40/β-propeller domain (repeats of approximately 40 amino acids terminating in tryptophan-aspartic acid (W-D) dipeptide). VPS54–GFP stably expressed in human neuroglioma cells pulled down TSSC1. Confocal fluorescence microscopy further confirmed the colocalization of VPS54 with TSSC1. Dysfunction of TSSC1 disrupted the transport of the Shiga toxin B subunit to the TGN and transferrin receptor recycling to the plasma membrane. FRAP analysis of VPS54-GFP in TSSC1 KD cells suggested the role of TSSC1 in GARP membrane association and/or TGN localization [39].

#### 4.4.2. Vps1p

Yeast two-hybrid analysis demonstrated the interaction between yeast dynamin Vps1p and Vps51p. The 33 aa segment (99–132 aa) near the *C*-terminal end of Vps51p was found to be responsible for this interaction. Glu127 and Tyr129 in the *C*-terminus of Vps51p are highly conserved and are involved in interaction with Vps1p. Confocal microscopy further supported the colocalization of Vps1p and Vps51p. Loss of Vps1p reduced the colocalization of Vps51p and Tlg1p with the late Golgi [40]. Hence, Vps1p could function with the GARP tethering machinery for efficient tethering/fusion at the TGN [41].

#### 4.4.3. RNF41

RNF41 (Ring Finger Protein 41) is an E3 ubiquitin ligase involved in the intracellular sorting and function of a diverse set of substrates. VPS52 has been shown to interact with RNF41. Through interaction via their coiled-coil domains, RNF41 ubiquitinates and relocates VPS52 away from VPS53 towards so-called RNF41 bodies [42].

#### 4.4.4. LRRK2

In mammalian cells, VPS52 has been shown to interact with LRRK2, serine/threonine-protein kinase, which phosphorylates a broad range of proteins involved in vesicle trafficking, autophagy, and neuronal plasticity [43,44,45]. The exact cellular function of LRRK2 is not well known, but mutation in LRRK2 (Gly2019Ser) is associated with Parkinson’s disease. LRRK2 mutant in *C. elegans* showed age-dependent loss of dopaminergic neurons. KD of vps-54 but not vps-50 in mutant worms exacerbated LRRK2-related neuronal toxicity, indicating that this defect is associated with the GARP complex [46]. Using a combination of proteomics and imaging techniques, a study in HEK293T cells reported the function of LRRK2 in mediating endosome-TGN transport by scaffolding the GARP–SNARE interaction (VAMP4; STX6), suggesting a connection between GARP and LRRK2 in Parkinson’s disease. In support of this proposed connection, siRNA-mediated VPS54 KD showed a decrease in the viability of dopamine neurons [46].

The known protein partners of GARP subunits are summarized in Table 1.

## 5. Functions of the GARP Complex

Although all GARP subunits are dispensable for life in yeasts [1], deletion of GARP causes an increased protein secretion, abnormal autophagy, and defects in vacuolar morphology [4,5]. GARP deletion also causes infertility in worms [54], a shorter life span in flies [53], and embryonic lethality in mice [55]. In this section, we have summarized the major described roles of GARP proteins in cell physiology.

### 5.1. GARP as a Molecular Tether for the Endosomal-Derived Vesicles

One of the most studied functions of the GARP complex is its role in retrograde vesicular transport from endosomes to TGN, which enables the retrieval of recycling transmembrane proteins like vacuolar protein sorting receptor Vps10p in *S. cerevisiae*. Vps10p binds to the pro-carboxypeptidase (Pro-CPY) in the TGN. Pro-CPY is one of the vacuolar hydrolase precursors that is delivered to the vacuole, while the receptor Vps10p needs to be recycled back to the TGN for further rounds of sorting. A mutation in any GARP subunit prevents Vps10p recycling to the TGN. Instead, Vps10p in GARP mutants is missorted to the vacuole and degraded. As a consequence of Vps10p degradation, GARP mutants missort precursors of vacuolar hydrolases, causing their secretion. Similarly, the recycling of TGN protein Kex2p and plasma membrane v-SNARE Snc1p is partially blocked in GARP mutants, resulting in their vacuolar degradation [4].

In mammalian cells, cation-independent mannose 6-phosphate receptor (CI-MPR/IGF2R/MPRI) performs a function similar to Vps10p in yeast. Knockdown of mammalian GARP subunits prevented the retrieval of CI-MPR to the TGN, resulting in missorting and secretion of lysosomal hydrolases. Blockade in the sorting of lysosomal hydrolases developed enlarged lysosomes as a result of the accumulation of undegraded materials in the lumen of the organelle [22].

There is also a blockage in the retrieval of TGN-localized recycling protein-TGN46 in GARP mutants [28]. Moreover, the recycling Golgi resident proteins (GPP130, TMEM165, and TGN46) are significantly depleted in VPS54KO and VPS53KO cells [56]. Similarly, the transport of the Shiga toxin is inhibited in the cells with siRNA-mediated KD of GARP subunits [22].

Based on GARP mutant phenotypes and on the similarity between GARP and other CATCHR complexes, it was proposed that GARP serves as a TGN-located molecular tether for the recycling of endosome-derived vesicles (Figure 3). In support of this model, yeast vps54 mutant cells accumulate multiple vesicular structures [2]. In addition, siRNA depletion of GARP subunits in HeLa cells caused redistribution of recycled CI-MPR to a vesicle-like “haze” [22]. It will be essential to isolate and characterize GARP-dependent membrane carriers and investigate the interplay between GARP and other players of the TGN vesicle tethering/docking/fusion machinery.

### 5.2. GARP as a Regulator of SNARE Complexes

Like other MTC vesicular tethers [30,57,58,59,60] the GARP complex physically interacts with a specific subset of SNARE molecules. Mammalian GARP complex interacts with endosome-TGN STX16 SNARE complex (STX16/STX6/VTI1A/VAMP4), and GARP KD cause decreased formation and/or stability of this complex [28]. In addition, GARP has been shown to regulate the localization and stability of intra-Golgi Qb-SNARE GOSR1 and Qc SNARE BET1L. These v-SNAREs work in the STX5/GOSR1/BET1L/YKT6 SNARE complex to facilitate the fusion of intra-Golgi recycling vesicles. Immunofluorescence microscopy demonstrated the depletion of Golgi GOSR1 and BET1L signals in GARP-KOs. Consistently, the total protein abundance of v-SNAREs in the GARP-KOs was significantly decreased [61]. This indicates that the GARP is involved in regulating of at least two different SNARE complexes promoting the fusion of cargo vesicles not only in TGN but also in the early-Golgi compartments.

### 5.3. Role of the GARP Complex in the Maintenance of Golgi Glycosylation Machinery

Glycosylation of proteins and lipids contributes to a variety of biological processes, and abnormal glycosylation in humans is a hallmark of many diseases. Mutations in proteins that are directly or indirectly involved in the glycosylation process can cause congenital disorders of glycosylation [62,63]. The glycosylation machinery is localized in the ER and Golgi [64], and therefore it was initially assumed that post-Golgi trafficking does not significantly contribute to the glycosylation process. Mutations in the GARP complex and GARP-associated clinical cases of glycosylation disorders have not been widely known. However, a recent publication described a 6-year-old patient with a neurodevelopmental disorder having a mutation in the VPS51 subunit of GARP/EARP complex and an abnormal pattern of glycosylation [65].

Similarly, the study in our lab revealed that KOs of GARP complex subunits VPS53 and VPS54 in different types of tissue culture cells (hTERT–RPE1, HEK293T and HeLa) [66] resulted in severe defects in both *N*- and *O*-protein glycosylation. Golgi glycosylation enzymes (MGAT1, B4GALT1, and ST6GAL1) were also significantly depleted. Retention using selective hooks (RUSH) assay showed that B4GALT1 was not retained at the Golgi complex in GARP-KO cells but was missorted to the endolysosomal compartment. Another *trans*-Golgi enzyme, ST6GAL1, was also not retained in the Golgi in GARP-impaired cells. It was shown that the defect in Golgi enzymes retention/recycling depends entirely on GARP and not on the EARP complex. The malfunction, mislocalization, and reduced stabilization of Golgi enzymes associated with a glycosylation defect were found to be rescued upon expression of the missing GARP subunit. Taken together, we concluded that GARP plays a crucial role in normal Golgi glycosylation by mediating the maintenance of the Golgi glycosylation machinery [56].

### 5.4. Role of the GARP Complex in Normal Golgi Physiology

Although the majority of GARP-sensitive proteins are predicted to reside in *trans*-Golgi compartments [56], the proteomic analysis of Golgi-enriched membranes in hTERT-RPE1 cells lacking GARP subunits revealed significant depletion of *cis*/*medial* Golgi proteins such as GLG1/MG-160, GALNT1, and MAN1A2 [61]. Recent results from our lab revealed that in human GARP KO cells, COPI vesicular coat complex, which is involved in intra-Golgi and Golgi-ER retrograde trafficking [67], is partially mislocalized to the ERGIC compartment. In GARP KO cells, the COPI accessory proteins GOLPH3, ARFGAP1, and GBF1 were displaced from the membrane, while BIG1 was relocated to the endolysosomal compartment. In addition, transmission electron microscopy revealed dysmorphic Golgi features in GARP mutants. In VPS54 KO cells, there is an enlargement of the Golgi structure, indicating an increase in the distance between the cisternae. In VPS53 KO cells, the Golgi structure was round, swollen, and disrupted [61]. This suggests that GARP dysfunction impacts not just *trans*-Golgi compartments but also the entire Golgi complex (Figure 4). Therefore, the GARP activity is required for normal Golgi physiology.

### 5.5. GARP and Lipid Homeostasis

Fröhlich et al. identified a crucial role for the yeast GARP complex in lipid homeostasis. Deletion of any component of the GARP complex resulted in the buildup of sphingolipid intermediates and long-chain bases, whereas pharmacological inhibition of sphingolipids can repair all observed GARP-KO mutant abnormalities. This shows that sphingolipid accumulation is the problem-causing factor in GARP-deficient cells [68]. Maintenance of the phospholipid’s asymmetric distribution from the outer to the inner leaflet of the plasma membrane (PM) is done by phospholipid transporter flippases. The mislocalization of PM localized flippases results in changes in lipid homeostasis [69]. Mutation in the GARP in yeast showed hypersensitivity to the phosphatidylethanolamine (PE) binding lantibiotic Ro by preventing cell growth. GARP mutants showed the localization of Dnf1p and Dnf2p to the vacuole, while in WT, it is localized to the PM. Vacuolar proteomics and lipidomics demonstrate missorting of flippases Dnf1p and Dnf2p, suggesting that remodeling of the lipid composition is one of the first occurring defects in GARP-depleted cells [70]. A study in flies showed that the GARP complex is involved in sterol transport, and knockout of Vps54 resulted in reduced lifespan and impaired arborization of *Drosophila melanogaster*. Significantly, the expression of a wild type copy of Vps54 in Vps54 KO flies rescued these mutant phenotypes [71].

The GARP complex is also involved as a regulator in the morphogenesis of *Candida albicans*. A peculiar feature of *Candida albicans* is the switch between yeast and filamentous hyphal form in response to cues. A genomic screen in *Candida albicans* revealed that deletion of GARP subunits impaired filamentation in response to Hsp90 inhibition. This filamentation defect was found to be due to the disruption of lipid homeostasis in the GARP mutant. GARP mutants have an increase in the number and size of lipid droplets compared to the control [72].

Dysfunction of the GARP complex resulted in missorting of lysosome resident protein NPC2, responsible for the egress of LDL-derived cholesterol from lysosomes. As a result, GARP-depleted cells have an accumulation of cholesterol. This phenotype is similar to Niemann–Pick type C disease, which is a neurodegenerative disease characterized by a massive accumulation of free cholesterol and other lipids in almost all cells throughout the body, particularly in the brain and liver [73].

### 5.6. Role of GARP Complex in the Secretory Pathway

Besides the endosomal-TGN trafficking function of the GARP complex and the promotion of SNARE function in membrane fusion, VPS52 has also been implicated in the regulation of the secretory pathway. The secretion of soluble secretory protein ss-sfGFP (super-folder GFP fused to signal sequence) was significantly decreased in VPS52 KO MDCK cells with subsequent accumulation of ss-sfGFP in lysosomes. This phenotype was prevented by the stable expression of VPS52. The functional importance of the *C*-terminal region of VPS52 for restoring secretory phenotypes, as well as for the sorting of lysosomal proteins, was assessed where amino acids 411 to 556 are required for ss-sfGFP sorting and secretion while amino acids 411 to 723 are required for efficient lysosomal protein sorting [74].

KO of GARP subunits in HEK293T cells caused severely defective anterograde transport of both glycosylphosphatidylinositol (GPI)-anchored and transmembrane proteins from the TGN [75]. Overexpression of VAMP4, TMEM87A or its close homolog TMEM87B in VPS54 KO cells partially restored endosome-to-TGN retrograde transport and anterograde transport of GPI-anchored proteins. The authors proposed that GARP- and VAMP4-dependent endosome-to-TGN retrograde transport is required for the recycling of molecules critical for efficient post-Golgi anterograde transport of cell-surface integral membrane proteins.

### 5.7. Hijacking of GARP by Intracellular Pathogens

To determine the host components essential for monkeypox virus infection, genome-wide insertional mutagenesis was done in HAP1 human cells. The screening identified the significance of the GARP complex in the generation of extracellular viruses. Extracellular viruses are responsible for cell-to-cell and long-distance virus transmission, both of which are needed for pathogenicity. In GARP KO (VPS52 KO and VPS54 KO) cells, the virus yield was drastically reduced, whereas the structure of the mature virion was unaffected. Rescuing the knockout with the expression of the WT copy of the affected GARP subunit restored the virus yield. Interestingly, the electron microscopy revealed a decrease in the membrane wrapping in GARP KO cells, indicating that GARP could be used for efficient membrane wrapping of the mature virion. In addition, confocal microscopy demonstrated that the GARP KO infected cell lacks actin tail production compared to controls which is one of the mechanisms for viruses’ escape. The role of Golgi trafficking machinery in the production of monkeypox and vaccinia virus is further supported by another study in COG KO cells where COG KO cells produced lower virus yield and possessed a reduced number of actin tails compared to WT [76,77].

## 6. GARP Complex Mutations and Pathogenesis

### 6.1. GARP Complex Mutations

Mutations in GARP subunits can disrupt the normal function of the GARP and lead to pathogenesis in humans, mice, and plants.

#### 6.1.1. Vps54 Null Mutant

The Vps54 null mutation is embryonically lethal in mice. On embryonic day E11.5, wild type embryos survived, while Vps54 null mutant embryos did not. The E11.5 embryos that had been sectioned exhibited spinal cord underdevelopment and nearly absent dorsal root ganglia. In addition, the atrial and ventricular myocardium exhibited significant hypoplasia, demonstrating the significance of Vps54 in embryonic development [55]. The null mutant of the Vps54 in *Caenorhabditis elegans* is either infertile or non-viable (allele tm585; Wormbase). Moreover, in *Drosophila melanogaster*, null mutants of the Vps54 homolog scattered (scat) have defective spermatogenesis [54].

#### 6.1.2. Wobbler Mouse

The wobbler mouse derived its name from its characteristically unsteady gait almost 66 years ago. In 1956, Falconer used the name “wobbler” to characterize the C57BL/Fa inbred mice with an unsteady walk due to a spontaneous mutation. Characterization of the phenotype revealed that wobbler mice exhibit muscular weakness, atrophy, and contractures, primarily in the forelimbs, head, and neck, as well as progressive motor neuron degeneration. This loss of motor neurons in the wobbler mouse’s brainstem and spinal cord mimics the human disease amyotrophic lateral sclerosis (AML). Therefore, the wobbler mouse has been widely used as an animal model for human neurodegenerative disorders [78,79]. The chromosomal mapping of the wobbler locus revealed that the disease-causing gene was autosomal, recessive, and presumably unique. The wobbler gene was found in proximal mouse chromosome 11 [80,81], which is homologous to human chromosome 2p13-14 [82]. This mutation was identified using positional cloning as a point mutation in the final exon of the VPS54 gene. In exon 23 of VPS54, wr/wr genomic DNA contains an A-T transversion in the second position of codon 967, leading to the substitution of hydrophobic leucine for hydrophilic glutamine. This hydrophobic-to-hydrophilic substitution destabilizes the tertiary structure, resulting in a decrease in VPS54 concentration. As a result, levels of the mutant protein are drastically reduced in all tissues of the wobbler mouse, including the spinal cord, which contains numerous motor neurons [83]. Similarly, the levels of VPS53 are also decreased due to the degradation of the excess, unassembled protein and as a result of a reduction in the GARP complex. This wobbler phenotype is specifically due to the decrease in VPS54 stability, as it was rescued by the development of BAC transgenic (created by microinjection of fertilized eggs) mice carrying the wild type VPS54 subunit [55].

In addition, the wobbler mouse exhibited a deficiency in spermatogenesis. The wobbler mutant’s spermatogenesis deficiency is related to an anomaly comparable to human globozoospermia. In human globozoospermia, round-headed spermatozoa lack an acrosome, preventing them from interacting normally with oocytes. Using mass spectrometry, the testis of wobblers was shown to have altered proteins compared to those of the wild type. FABP3 was identified as the protein with the greatest decrease in wobbler testes, but HSP70 and HSP90 were considerably enhanced [84], indicating that VPS54 plays an essential role in infertility. This defect of the wobbler mouse was also corrected in transgenic mice containing WT VPS54, as the transgenic mouse generated viable progeny in multiple matings. The testes also showed a normal number of flat sperm heads and rescued wobbler phenotype [55].

### 6.2. Pathogenesis of GARP Mutations

VPS51 is essential for normal brain development. Whole exome sequencing of two siblings (one was 9-year-old and another 30 months) demonstrating delayed psychomotor development, absent speech, severe intellectual disability, and postnatal microcephaly, with cerebellar atrophy in the 9-year-old and hypoplastic corpus callosum in the 30 month old younger sister, revealed a variant in the VPS51 gene (reference sequence: NM_013265.3) consisting of an in-frame deletion of a CTT codon in exon 5 (c.1419_1421del; p (Phe474del)) detected in the homozygous state in both sisters [85]. Similarly, in another study, a 6-year-old patient demonstrating severe global developmental delay and dysmorphic features has been reported with compound heterozygous mutations (c.1468C > T/p.Asp745Thrfs*93 and c.2232delC/p.Arg490Cys) in VPS51 [65]. The mutation in one allele causes a frameshift that produces a long but highly unstable protein that is degraded by the proteasome. In contrast, the other mutant allele produces a protein with a single amino acid substitution that is stable but assembles less efficiently with the other GARP/EARP subunits [65].

A homolog of VPS51 in the plant *Arabidopsis thaliana*, Unhinged UNH-1 is required for the leaf shape and vein pattern through the maintenance of the PIN1 polarity at the plasma membrane. PIN1 is a component of the auxin efflux carrier that regulates the direction of cell-to-cell auxin transport. PIN polarity is governed by subsequent endocytosis, transcytosis, and recycling back to the plasma membrane. In the epidermal cells of unh-1 leaf margins, PIN1 expression is increased. PIN1 mutants have pointed and narrower leaves with fewer secondary and higher-order veins and a lack of distal vein junctions compared to wild type leaves. This *unh-1* leaf phenotype is suppressed by the *pin1* mutation, confirming the notion that the phenotype is the consequence of increased PIN1 expression in the marginal epidermis. The study demonstrated that UNH-1 is important for reducing the expression of PIN1 within margin cells, possibly by targeting PIN1 to the lytic vacuole. Similar to yeast, the UNH-1 is required for vacuolar trafficking and mutation of UNH-1 resulted in reduced vacuolar targeting and increased secretion of a vacuolar targeted fluorophore [47]. Hence, VPS51 is essential for the neurodevelopment in animals and normal morphology of plant leaves and veins.

VPS52 has been demonstrated to play a role in cancer biology. In gastric cancer cell lines, there is a lower basal expression of VPS52, in contrast to normal gastric epithelial cell lines. In gastric cancer cells in vitro, there is a decrease in cell viability through the induction of apoptosis. Overexpression of VPS52 in vivo has decreased tumor weight and volume. Loss of heterozygosity (9/17 samples) and a stop-gain mutation of VPS52 were discovered in the tissues of gastric cancer. Hence, VPS52 has been shown to be a tumor suppressor [86].

VPS52 has been shown to play a crucial role in protecting the plants (*Arabidopsis* and *Potato*) from aphid infestation. In these species, aphid effector Mp1 associates with VPS52 and reduces the level of plant VPS52. At the same time, an increase in the expression of VPS52 showed colocalization of Mp1 to the prevacuolar compartments. High VPS52 levels in plants negatively impact virulence and act as an important virulence target [87].

Mutation in the VPS53 subunit of the GARP complex was shown to be involved in progressive cerebello-cerebral atrophy 2 (PCCA2). The common features of PCCA2 are profound mental retardation, progressive microcephaly, spasticity, and early onset epilepsy. Whole exome sequencing identified c.2084A > G p. (Gln695Arg) missense mutation in exon 19 that replaces a neutral amino acid with a positively charged one within the second helix of the surface of VPS53′s conserved *C*-terminal. Similarly, the c.1556 + 5G > A splice site mutation in the fifth nucleotide of intron 14, generates an unstable transcript that is predicted to encode a truncated protein, leaving the affected individuals with no functional VPS53 *C*-terminus and partially dysfunctional GARP [88]. Similarly, whole exome sequencing of two female siblings affected with complicated Hereditary spastic paraparesis revealed a mutation in the VPS53 subunit of the GARP complex (c.2084A > G: c.2084A > G, p.Gln695Arg) [89].

VPS53 has been shown to play a pivotal role in suppressing the malignant properties of colorectal cancer (CRC). In CRC tissues, the protein level of VPS53 was significantly decreased compared to normal tissues. VPS53 in CRC tissues showed a positive correlation with autophagy protein Beclin 1, indicating that autophagy induction can facilitate CRC cell death. Similarly, inhibition of autophagy by Inhb attenuated the suppression of proliferation, invasion, and migration of cancer cells. Hence, VPS53 has a crucial role in the regulation of CRC progression. Moreover, overexpression of VPS53 in CRC cells has been shown to inhibit cell growth by about 50% [90]. This inhibitory effect was associated with the apoptotic process, including DNA fragmentation, the release of cytochrome c from mitochondria, and activation of caspase-9 and caspase-3 [91]. Similarly, VPS53 has been shown to be downregulated in other types of cancers, such as medullary thyroid cancer [92] and prostrate [93,94].

In a recent study, a Chinese herbal medicine GZFLW decreased the Y14 cell viability, thereby promoting apoptosis. A high-throughput strategy was employed to identify the differential expression level of proteins with or without GZFLW treatment. There was a difference in VPS53 expression level between the GZFLW treated group and the control group. The VPS53 expression level was 1.7 times higher in the GZFLW treated group. Hence, VPS53 was identified as a potential target for the treatment of endometriosis [95].

## 7. Future Perspectives in GARP Studies

GARP was initially described as a protein complex essential for the sorting of vacuolar hydrolases and recycling of membrane proteins from the endosomal/prevacuolar compartment to TGN. More than twenty years after the discovery of the GARP complex, there is still much unknown about this evolutionarily conserved machinery. Mutant phenotypes in cells lacking GARP subunits go beyond the defects in endosome–TGN trafficking and include glycosylation, intra-Golgi, and secretory defects. What is the immediate impact of GARP dysfunction? What are the secondary effects resulting from a prolonged depletion of GARP machinery? Although the GARP structurally belongs to the CATCHR family of vesicle tethering complexes like COG, Dsl1, and exocyst [14,19], it is not known what specific functions GARP performs. Direct evidence for GARP capturing specific transport vesicles is still lacking. Does it capture vesicles independently or together with coiled-coil tethers? Does it capture only one kind of transport vesicle? What specific protein and lipid cargo depends on the GARP complex? It will be essential to isolate and further characterize GARP-dependent membrane carriers and investigate the interplay between GARP and other players of TGN vesicle tethering/docking/fusion machinery. GARP is proposed to promote the assembly of TGN SNAREs, but the exact mode of GARP–SNARE interaction is unclear. Does GARP interact only with the STX16 SNARE complex? Does it bind and regulate STX16-interacting SM protein VPS45? Furthermore, GARP is required not only for retrograde trafficking, it also somehow regulates post-Golgi anterograde trafficking [75] and sphingolipid homeostasis [68,70].

To understand the function of the GARP complex, it is equally essential to elucidate the dynamic nature of the GARP structure. Little is known regarding either the overall or subunit architecture of the mammalian GARP complex. Is it the same for the free and membrane-bound forms of the complex? It is likely that GARP is switching from the “close” conformation to the “open” one to tether inbound transport intermediates. GARP and EARP complexes are known to share the VPS51/52/53 trimer. The GARP complex is proposed to localize in the TGN, while EARP is on the endosomes. Currently, it is not clear how a difference in one subunit between the GARP and EARP complex is translated into localization and function of the whole complex. It is also not clear if the shared VPS51/52/53 trimer is dynamically switching between GARP and EARP conformations or, alternatively, if GARP and EARP complexes are stable and never exchange their subunits. It is also necessary to identify the additional factors (like SNAREs, coiled-coil tethers, and small GTPases) that make GARP unique from the EARP complex.

The GARP complex is evolutionarily conserved, and its dysfunction can lead to pathogenesis in humans, mice, and plants. GARP deletions are not essential on the cellular level, but in mice, VPS54 null causes embryonic lethality. How can the dysfunction of a GARP be embryonically lethal? What are the specific functions of GARP in different cell types? Why does the GARP deficiency primarily affect neuronal function? It is also necessary to dig deeper into the signaling pathways, as GARP is associated with many cancers [90,91,92,93,94,95]. Initial GARP studies were performed with yeast and several mammalian tissue culture cell models, but it is clearly essential to study the GARP complex in other human tissues and other species. Therefore, it is necessary to connect the dots and determine GARP’s role in Golgi physiology that eventually contributes to neurodevelopmental and other human disorders.

GARP is essential for Golgi physiology. It is involved in many cellular processes and has been implicated in the pathophysiology of a number of disorders. Although the GARP complex has been the subject of intensive investigations since its discovery twenty years ago, it is abundantly clear that future detailed studies are necessary to fully understand the role of the GARP complex in physiology of the cell.

## Figures and Tables

**Figure 2 ijms-24-06069-f002:**
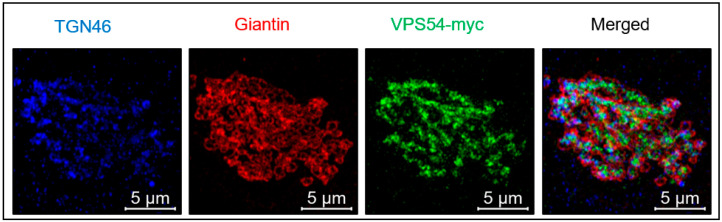
GARP is localized to *trans*-Golgi/TGN in human cells. hTERT–RPE1 VPS54 knock out cells stably expressing mVPS54–13myc were stained with antibodies to TGN protein TGN46, *cis*/*medial* Golgi protein Giantin and myc (VPS54). Airyscan super-resolution images (MIPs of 14 z-stack) were collected with 63x oil 1.4 NA objective on Zeiss LSM880.

**Figure 3 ijms-24-06069-f003:**
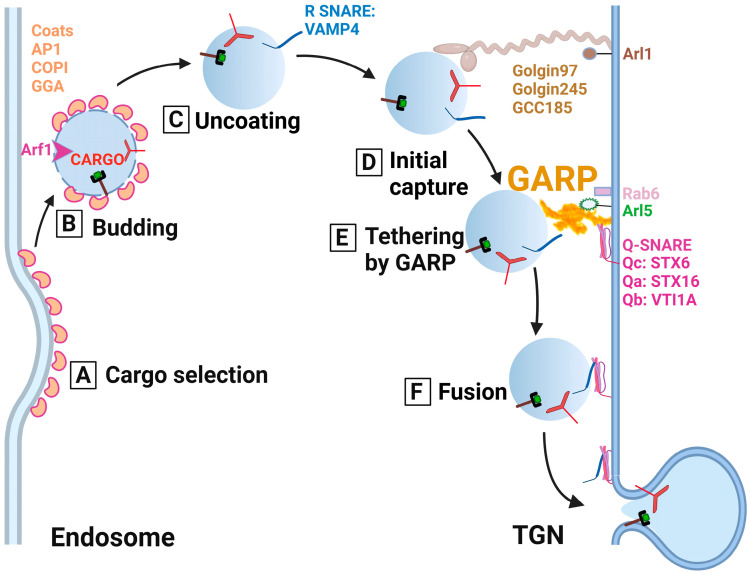
Cartoon depicting the predicted role of the GARP complex and other trafficking components in the tethering of endosome-derived vesicles to the TGN. (A) Cargo selection: Recycling soluble and transmembrane cargo molecules such as the mannose-6-phosphate receptor and TGN46 are selected and packaged in the trafficking intermediates (vesicle) using ARF1 GTPase and a vesicular coat (AP1, GGA or COPI). (B) Budding: After the vesicle is formed, it is budded off the endosomal compartment. (C) Uncoating: ARF1 hydrolyzes GTP, and the coat gradually falls off. (D) Initial capture: Once the vesicle reaches proximity to the TGN, coiled-coil tethers (Golgin97, Golgin245, and GCC185) perform the initial capture. (E) Tethering by GARP: GARP complex binds coiled-coil tethers and SNAREs to control and coordinate docking of the vesicle with the TGN membrane. (F) Fusion: Following the tethering by the GARP complex, the *trans*-SNARE complex is formed, which results in vesicle fusion with the TGN to deliver the recycling molecules.

**Figure 4 ijms-24-06069-f004:**
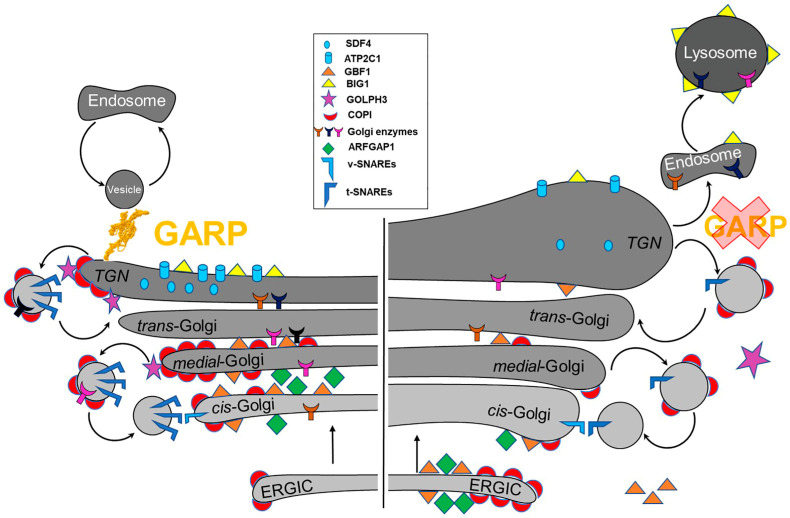
GARP dysfunction results in the depletion and mislocalization of multiple Golgi proteins, altering Golgi homeostasis. (**Left**) A schematic of Golgi proteins in wild type cells. (**Right**) GARP deficient cells are depleted for multiple resident Golgi proteins. COPI coats and ARFGAP1 are mislocalized to ERGIC., while BIG1 is relocated to endolysosomal compartments.

**Table 1 ijms-24-06069-t001:** GARP subunits and their proteins partners in different model organisms.

Gene Name	Organisms	Common Names	Length of the Protein	Protein Partners				References
				SNAREs	GTPases	EIPR1/ TSSC1	Others	
**VPS51**	Yeast	Vps51/Vps67/Whi6/Api3/YKR020W	164	Tlg1p [3,4,27]			Vps1 [40]	
	Plant	VPS51, UNH	780					[47]
	Worm	vps-51, B0414.8	700					[48]
	Fly	Vps51	740		Arl5 [26]			
	Fish	vps51, ffr	827					
	Human:	VPS51, ANG2/C11orf2, C11orf3	782	STX6 [18,49]		[39]		[6]
**VPS52**	Yeast	VPS52/Sac2/YDR484W, D8035.27	641		Ypt6 [24]			
	Plant	VPS52, POK, TTD8, At1g71270, F3I17.8	707					[50]
	Worm	vps-52	702		Rab6 [48]			
	Fly	Vps52	662		Arl5 [26]			
	Fish	Vps52	724					
	Human	VPS52, SACM2L	723	STX6 [46]; STX16; VAMP4; VTI1A [28]; STX10 [23]	Rab6 [23] Arf6 [37]		RNF41 [42]	
**VPS53**	Yeast	VPS53, YJL029C, J1258	822		Arl1 [36]			
	Plant	VPS53, HIT1, At1g50500, F11F12.15, F17J6.4	828					
	Worm	vps-53	798					
	Fly	Vps53	683					
	Fish	vps53	831					
	Human	VPS53, PP13624	832	STX6; STX16; VAMP4; VTI1A [28]				
**VPS54**	Yeast	VPS54, CGP1, LUV1, RKI1, TCS3, YDR027C, PZF889, YD9813.05C	889					[51]
	Plant	VPS54, At4g19490, F24J7.50	1034					
	Worm	vps-54, T21C9.2	1058					
	Fly	scat, CG3766	940		Rab5,7,11 [52]			[53]
	Fish	vps54	998					
	Human	VPS54, HCC8	977	STX6; STX16; VAMP4; VTI1A [28]				

Yeast: *Saccharomyces cerevisiae*; Plant: *Arabidopsis thaliana*; Worm: *Caenorhabditis elegans*; Fly: *Drosophila melanogaster*; Fish: *Danio rerio*; Human: *Homo sapiens*.

## Data Availability

Not applicable.

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
