# Peer review of "Role of GARP Vesicle Tethering Complex in Golgi Physiology"

_ijms, 2023, doi:10.3390/ijms24076069_

Round 1

Reviewer 1 Report

Overall, this article provides a concise summary of the current understanding of the GARP complex and its role in Golgi physiology. The authors effectively introduce the GARP complex as a member of the CATCHR family and describe its proposed function in tethering and promoting the fusion of endosome-derived trafficking intermediates.

The article also highlights recent studies that have identified novel roles for GARP in Golgi physiology and have linked mutations in GARP subunits to cellular defects. This information provides insight into the importance of GARP in cellular processes and the potential consequences of dysfunction.

Author Response

We appreciate positive comments on our manuscript. The manuscript was significantly updated to reflect the reviewer's comments. We hope the updated manuscript is now suitable for publication in the Journal.

Reviewer 2 Report

1.      Lines 38-45, discovery of subunit-51 brought the name GARP-complex but the mutation in the respective subunit does not affect the assembly of the complex, so is its functions such defects in vacuole morphology or SNARE-trafficking are independent of the functions of GARP-complex? Authors are suggested to clearly mention their view point in these lines.

2.      Line 51-52; human VPS51 subunit have more than 700 aa; so, does author still wish to consider this as an exception? If not then please update the manuscript accordingly.

3.      Fig.1. for alphafold structures please include the date of retrieval in the legend.

4.      If possible, authors are suggested to realize the 3d assembly of both GARP & EARP in their review or else try to minimalize the EARP part of their review.

5.      Table.1. why STX6 is not included as an interaction partner while same is described by the authors in lines 257-61; please rectify.

6.      It is advised that for Fig.3. authors specifically mention which cargos they are referring to because it may be different than the cargos at other membranous cellular sites.

7.      Authors are advised to double check the information mentioned in lines 426-28, from the perspective of the research model used. In fact, if exists any, authors could explore and include, the GARP functional diversity across the used cell models.

8.      Authors should consider improving the clarity of their figures.

9.      Please revisit the manuscript for other format related, grammatical or typos related issues.

Author Response

We appreciate positive comments on our manuscript. The manuscript was significantly updated to reflect the reviewer's comments. We hope the revised manuscript is now suitable for publication in the Journal.

Point-by-point reply:

  1. Lines 38-45, discovery of subunit-51 brought the name GARP-complex but the mutation in the respective subunit does not affect the assembly of the complex, so is its functions such defects in vacuole morphology or SNARE-trafficking are independent of the functions of GARP-complex? Authors are suggested to clearly mention their view point in these lines.

We acknowledge the reviewer for pointing this out. We have now modified the text to clarify the role of Vps51 in the GARP complex. Lines 40-49.

“This newly identified subunit of the complex was equally crucial to the membrane trafficking as other subunits. The yeast cells that lack Vps51p also have defects in vacuole morphology and recycling of SNARE Snc1p to the plasma membrane [4]. Similarly, in human cells, VPS51 knock-down (KD) partially dispersed the endosome to trans-Golgi network (TGN) recycling proteins CI-MPR and TGN46. The uptaken Shiga toxin did not reach the Golgi in VPS51 KD cells, but scattered throughout the cytoplasm [6]. This indicates that Vps51 has a similar function to that of VFT complex. At the same time, the assembly of the Vps52p/53p/54p complex can still take place in vps51 mutant even though the level of these subunits was highly reduced compared to WT cells [1, 3].”

  1. Conibear, E. and T.H. Stevens, Vps52p, Vps53p, and Vps54p form a novel multisubunit complex required for protein sorting at the yeast late Golgi. Molecular biology of the cell, 2000. 11(1): p. 305-323.
  2. Siniossoglou, S. and H.R. Pelham, Vps51p links the VFT complex to the SNARE Tlg1p. Journal of Biological Chemistry, 2002. 277(50): p. 48318-48324.
  3. Conibear, E., J.N. Cleck, and T.H. Stevens, Vps51p mediates the association of the GARP (Vps52/53/54) complex with the late Golgi t-SNARE Tlg1p. Molecular biology of the cell, 2003. 14(4): p. 1610-1623.
  4. Perez-Victoria, F.J., et al., Ang2/fat-free is a conserved subunit of the Golgi-associated retrograde protein complex. Mol Biol Cell, 2010. 21(19): p. 3386-95.

  1. Line 51-52; human VPS51 subunit have more than 700 aa; so, does author still wish to consider this as an exception? If not then please update the manuscript accordingly.

Thank you for making this point. Please refer to the changes made in lines 56-59

“In most eukaryotes, the length of the GARP complex subunits ranges from 650 to 1700 amino acid residues whereas yeast Vps51p subunit is an exception as it is only 164 amino acids long [10].”

Reference

  1. Bonifacino, J.S. and A. Hierro, Transport according to GARP: receiving retrograde cargo at the trans-Golgi network. Trends in cell biology, 2011. 21(3): p. 159-167.

3. Fig.1. for alphafold structures please include the date of retrieval in the legend.

We have now added the date of retrieval in the legend. Please see the updated review line 118-121.

“Predicted 3D structures of individual VPS51-54 human proteins were updated in AlphaFold DB version 2022-11-01, created with the AlphaFold Monomer v2.0 pipeline. The predicted 3D structure of GARP complex was assembled on 12.27.2022 and colored using ChimeraX.”

  1. If possible, authors are suggested to realize the 3d assembly of both GARP & EARP in their review or else try to minimalize the EARP part of their review.

Thank you for the suggestion. We agree with your point and removed predicted 3D structure of EARP from the review.

  1. Table.1. why STX6 is not included as an interaction partner while same is described by the authors in lines 257-61; please rectify.

We have now included STX6 in the table.

  1. It is advised that for Fig.3. authors specifically mention which cargos they are referring to because it may be different than the cargos at other membranous cellular sites.

We acknowledge the reviewer for this comment. We have added “transmembrane cargo molecules such as mannose-6-phosphate receptor (M6PR) and TGN46” in line 275-276.

  1. Authors are advised to double check the information mentioned in lines 426-28, from the perspective of the research model used. In fact, if exists any, authors could explore and include, the GARP functional diversity across the used cell models.

Thank you for the comment. We have checked the information and re-wrote the paragraph. Please, refer to line 431-438.

“Interestingly, the electron microscopy revealed the decrease in the membrane wrapping in GARP-KO cells indicating that GARP could be used for efficient membrane wrapping of the mature virion. In addition, confocal microscopy demonstrated that GARP-KO infected cell lacks actin tail production compared to controls which is one of the mechanisms viruses escape. Role of Golgi trafficking machinery in the production of monkeypox and vaccinia virus is supported by another study in COG KO cells where COG KO cells produced lower virus yield and possess a reduced number of actin tails compared to WT [76][77].”

References

  1. Realegeno, S., et al., Conserved Oligomeric Golgi (COG) complex proteins facilitate orthopoxvirus entry, fu-sion and spread. Viruses, 2020. 12(7): p. 707.
  2. Realegeno, S., et al., Monkeypox virus host factor screen using haploid cells identifies essential role of GARP complex in extracellular virus formation. Journal of virology, 2017. 91(11): p. e00011-17.

8. Authors should consider improving the clarity of their figures.

The figures are enlarged with larger fonts in the updated review.

  1. Please revisit the manuscript for other format related, grammatical or typos related issues.

We have rechecked the manuscript and corrected grammatical and typos.

Reviewer 3 Report

In the review entitled “Role of GARP vesicle tethering complex in Golgi physiology”, the authors summarized the data obtained regarding different aspects of GARP complex, such as its composition, structure and localization, as well as GARP functions related to Golgi physiology and the cellular defects associated with the dysfunction of GARP subunits.

The text and data are presented in a rational way, making it easy to read and understand. The idea proposed in the abstract of this review article is original and exciting. Currently, there are no other review articles covering this topic. I’m very enthusiastic about this manuscript and I have no major concerns. I only have minor comments (please, see below).

Minor revisions

Line 53 – the expression “The amino-terminal amino-acid sequence” is not clear. Please, rewrite it or clarified it.

Line 262 – table1. The format of the table needs to be organized. References appear in different columns.

Line 430 – Please, increase the font size of the words in the figure or increase the size of the image.  

Author Response

We appreciate positive comments on our manuscript. The manuscript was significantly updated to reflect the reviewer's comments. We hope the revised manuscript is now suitable for publication in the Journal.

Point-by-point reply:

Line 53 – the expression “The amino-terminal amino-acid sequence” is not clear. Please, rewrite it or clarified it.

Thank you for the comment. We have now clarified the sentences as “The amino-acid sequence in the N-terminus of every identified subunit of the GARP”. Please refer line 60.

Line 262 – table1. The format of the table needs to be organized. References appear in different columns.

We have now re-organized the table and references.

Line 430 – Please, increase the font size of the words in the figure or increase the size of the image.  

We have now increased the font size of the words in the figure and also increased the size of the image.